# Role of Endoscopic Ultrasound in Liver Disease: Where Do We Stand?

**DOI:** 10.3390/diagnostics11112021

**Published:** 2021-10-31

**Authors:** Tajana Pavic, Ivana Mikolasevic, Dominik Kralj, Nina Blazevic, Anita Skrtic, Ivan Budimir, Ivan Lerotic, Davor Hrabar

**Affiliations:** 1Department of Gastroenterology and Hepatology, University Hospital Center Sestre Milosrdnice, 10000 Zagreb, Croatia; dominik.rex@gmail.com (D.K.); nina.blazevic05@gmail.com (N.B.); ivan.budimir.zg@gmail.com (I.B.); ivanlerotic@yahoo.com (I.L.); davor.hrabar@kbcsm.hr (D.H.); 2Department of Gastroenterology, University Hospital Center Rijeka, 51000 Rijeka, Croatia; ivana.mikolasevic@gmail.com; 3Department of Pathology, Merkur University Hospital, 10000 Zagreb, Croatia; skrtic.anita@gmail.com

**Keywords:** EUS, endoscopic ultrasound, chronic liver disease, hepatology

## Abstract

As the burden of liver disease in the general populace steadily increases, so does the need for both advanced diagnostic and treatment options. Endoscopic ultrasound is a reliable diagnostic and therapeutic method that has an established role, foremost in pancreatobiliary pathology. This paper aims to summarize the growing role of endoscopic ultrasound in hepatology based on the search of the current literature. A number of applications of endoscopic ultrasound are reviewed, including both noninvasive methods and tissue acquisition in focal and diffuse liver disease, portal hypertension measurement, detection and management of gastric and esophageal varices, treatment of focal liver lesions and staging of pancreatobiliary malignancies, treatment of cystic and solid liver lesions, as well as liver abscess drainage. Both hepatologists and endoscopists should be aware of the evolving role of endoscopic ultrasound in liver disease. The inherent invasive nature of endoscopic examination limits its use to a targeted population identified using noninvasive methods. Endoscopic ultrasound is one the most versatile methods in gastroenterology, allowing immediate access with detection, sampling, and treatment of digestive tract pathology. Further expansion of its use in hepatology is immanent.

## 1. Introduction

At the beginning of the 21st century, chronic liver disease (CLD) is a significant public health concern. It has been reported that globally 1.5 billion persons had CLD in 2017, most commonly resulting from non-alcoholic fatty liver disease (NAFLD) [1]. Moreover, the burden of NAFLD and alcohol-related liver disease (ALD) is expected to increase over the coming years [2]. Noninvasive diagnostic modalities developed in recent years have greatly advanced the evaluation of CLD, but there are still many clinical situations where accurate diagnosis and staging depend on histopathology. Endoscopic ultrasound (EUS) is an unavoidable method for the evaluation of the pancreatobiliary and upper gastrointestinal tract with an expanding role in the field of hepatology. The limitations of conventional diagnostic tools and percutaneous interventions for liver disease, mostly done with transabdominal ultrasound (US) or computed tomography (CT) guidance, have made EUS an attractive alternative, predominantly due to the enhanced imaging quality and safety profile along with the biopsy acquisition ability regardless of body habitus [3,4].

In this review, we summarized the current data on EUS applications in liver disease. A thorough search for studies published before 30 June 2021 was performed using the Medline/PubMed and Embase databases with the keywords “endosonography”, “endoscopic ultrasound”, “EUS”, “liver disease”, “hepatology”, “cirrhosis”, “portal hypertension”, “interventional EUS”, “therapeutic EUS”, “liver biopsy”, “fine needle aspiration” and “fine needle biopsy”. The search yielded 834 articles, 172 of which were included in this review.

## 2. EUS in Diffuse Liver Lesions

### 2.1. EUS Elastography

The presence of liver fibrosis is the most important prognostic factor in determining the liver-related and overall morbidity and mortality in CLD patients. Still, the “gold standard“ for fibrosis detection and grading is a liver biopsy (LB). However, due to the limitations of LB, over the last decade, a number of noninvasive methods for fibrosis detection have been investigated. Some of them are elastographic methods that were developed to overcome the limitations of LB. Generally, the reduced elastic rebound is highly suggestive for stiffer tissue. This finding, in the context of CLD, indicates liver fibrosis and/or cirrhosis or some other pathological process, such as malignant focal lesions [5,6,7,8]. Elastography modalities currently in use are transient elastography (FibroScan), 2-dimensional shear-wave elastography (SWE), acoustic radiation force impulse (ARFI) imaging, real-time elastography (RTE), and magnetic resonance (MR) imaging elastography. RTE is different from other ultrasound elastography techniques because it can measure heartbeat-induced strain, which allows relatively objective qualitative and semiquantitative results [5,6,7,8,9,10].

Elastographic methods, such as FibroScan, have shown a good correlation to histological findings [5,6,7,8,9]. However, there are some limitations in terms of FibroScan use, such as the applicability of its measurements with a transabdominal approach in obese patients, those with ascites, or in those with narrow intercostal spaces. In addition, FibroScan has lower applicability in discriminating between intermediate stages of fibrosis [6,7,8,9,10,11].

Additionally, transabdominal elastography is usually performed over the right lobe of the liver; therefore, the association of elastography findings with histological findings is affected by variability between the right and left lobes of the liver [6,7,8,9,10,11,12]. On the other hand, EUS processors have the capability to carry out elastography both in the right and left lobes of the liver. The use of EUS elastography is not limited by obesity (i.e., body mass index) or by ascites [8,9,10,11,12]. According to some authors, it is a reasonable idea that EUS RTE could be more sensitive than transabdominal RTE in terms of the liver fibrosis stage. This is mainly due to the shorter penetration depth in the EUS approach than in the transabdominal approach (thick abdominal wall vs. thin gastric wall) [10]. Moreover, during the EUS elastography examination, there is concomitant upper gastrointestinal tract luminal examination, incorporating both esophagogastroduodenoscopy (EGD) and elastography measurements into one procedure at the same time [8,9,10,11,12]. Recently, Schulman AR et al. [13] analyzed 50 patients that underwent EUS RTE, which was performed to synthesize the liver fibrosis index (LFI) in each patient. In the study, the patients were divided into normal liver (*n* = 26), fatty liver (*n* = 16), and cirrhosis groups (*n* = 8). Authors had found that LFI, computed from RTE images, correlates with abdominal imaging. In addition, it can distinguish normal, fatty, and cirrhotic-appearing livers [13]. Although the results of this study are interesting and promising and the authors showed that EUS RTE may be a good method for the noninvasive assessment of liver fibrosis in obese patients, it has some limitations, such as LB was not performed in all patients and the sample size. EUS RTE is a much more invasive method in comparison to transabdominal RTE and often requires sedation. Additional studies are warranted to investigate the efficacy of these two methods in comparison to LB, as well as to determine the cutoff values for each fibrosis stage [10]. A potential role of EUS RTE for serial monitoring of dynamic changes in liver fibrosis in CLD patients included in esophagogastric varices (EGV) endoscopic surveillance programs should be investigated [9]. The main advantage of EUS RTE is its ability to evaluate the presence of focal liver lesions, parenchymal liver abnormalities, and complications of portal hypertension. Thus, the use of this method could reduce the number of procedures when more than one organ requires evaluation [8]. One more interesting aspect of EUS elastography is the detection, differentiation, and characterization of focal liver lesions. In one study, where 39 liver tumors were analyzed using RTE performed by US or EUS. The sensitivity, specificity, and accuracy of differentiation of benign and malignant lesions were 92.5%, 88.8%, and 88.6%, respectively [14]. However, further investigations that will investigate the potential of EUS elastography in this field are needed.

### 2.2. EUS-Guided Liver Biopsy (EUS-LB)

In recent decades, there have been many advances in noninvasive diagnosis and investigation of liver diseases, but LB remains the best means for obtaining and clarifying the underlying pathology, determining the severity of liver damage, monitoring disease progression, or supporting research [15,16]. The most important issue regarding the procedure is to obtain an adequate liver specimen, which will allow detailed histopathological interpretation. The American Association for the Study of Liver Diseases guidelines suggests that adequate LB specimens contain a tissue core of at least 2–3 cm in length with the presence of more than 11 complete portal tracts [16]. Percutaneous LB continues to be the most utilized technique for histopathological assessment of liver tissue, which uses an image-guided approach (US or CT) to reduce the complication rate. Transjugular biopsy accesses the liver through the superior vena cava and the hepatic vein, without traversing the liver capsule, which is useful in patients with bleeding diathesis, presence of ascites and morbid obesity, or in those who could benefit from simultaneous direct measurement of the hepatic venous pressure gradient (HVPG) [17]. Major complications following traditional methods of LB reach up to 2.5%, the most common being hemorrhage and pain [18,19], while mortality rates occur at 0.2% [20]. Further, the heterogeneity of liver fibrosis may contribute to sampling variability, which has been recognized as a potential pitfall of standard LB techniques [16,21].

Since the first published cases of EUS-LB in 2007 using a novel Tru-Cut core biopsy needle (QuickCore; Cook Medical, Winston Salem, NC, USA) [22,23,24], there have been numerous studies showing comparable adequacy and complication rates [3]. Yet, the EUS technique still affords many advantages over percutaneous and transjugular approaches. Due to the proximity of the ultrasound device to the liver, EUS allows for a detailed view of a patient’s anatomy in real-time and the avoidance of other structures, including the adjacent vasculature and major bile ducts, thus reducing procedure-related complications [25]. In this way, multiple cores from both right and left liver lobes can be obtained, increasing the adequacy and yield of tissue [26]. Additionally, EUS-LB is performed with either conscious sedation or under anesthesia, significantly improving patient tolerance and comfort [27,28]. The procedure is quick, adding only a few minutes to the overall procedure time [29]. We often perform EUS for the evaluation of elevated liver enzymes in patients with a dilated common bile duct, and in case of non-diagnostic findings, patients can undergo EUS-LB in the same session, which is likely to reduce overall time, cost of multiple procedures, and expedite clinical management [26]. Finally, EUS-LB has a shorter average recovery time compared to conventional LB methods [28].

To date, several studies have evaluated the diagnostic yield, accuracy, and safety profile of EUS-LB in patients with CLDs of various causes (Table 1) [22,23,24,25,29,30,31,32,33,34,35,36,37,38,39,40,41,42,43,44,45,46,47,48].

The first meta-analysis performed on this topic included nine studies from 2009 to 2016, which demonstrated that EUS-LB has a similar diagnostic yield (93.9% (95% confidence interval [CI], 84.9–97.7)) and adverse event rates (2.3% (95% CI; 1.1–4.8); the pooled rate of bleeding 1.2%) when compared to data from studies of percutaneous and transjugular approaches [49]. The subgroup analyses based on the needle type (core needle (QuickCore and ProCore, Cook Medical, Bloomington, USA) vs. fine-needle aspiration (FNA) needle, all 19 G) showed that the FNA needle had a significantly lower rate of achieving insufficient specimens than core biopsy needles (4% vs. 20%, *p* = 0.03). The possible explanation for this lies in the fact that the majority of inadequate specimens were associated with the use of the QuickCore biopsy needle, which is no longer commercially available [50]. In one study comparing the QuickCore and Procore needles for EUS-LB, the QuickCore was significantly inferior in terms of obtaining a histologic diagnosis (73% vs. 97%), number of complete portal triads (CPT), and aggregate specimen length [32]. In the past several years, multiple dedicated EUS-guided fine-needle biopsy (FNB) devices with enhanced tip designs for maximal tissue acquisition have been made available for commercial use. Schulman et al. tested four EUS needle types against two 18G percutaneous needles on human cadaveric tissue (19G FNB SharkCore, Medtronic, Sunnyvale, CA, USA; 19G FNA Expect, Boston Scientific, Natick, MA, USA; 19G FNB Echo Tip HD ProCore, Cook Medical Inc., Bloomington, IN, USA; 22G FNB SharkCore, Medtronic, Sunnyvale, CA, USA) and reported that the novel 19G FNB needle was associated with the maximal number of CPTs. Moreover, a 22G FNB needle was not statistically different from an 18G percutaneous needle [51]. When comparing 22G FNB versus 19G FNA needles, tissue adequacy is higher for the 19G FNAs (88% vs. 68%, *p* = 0.03), mainly because samples obtained from a smaller caliber needle are more prone to fragmentation during specimen processing [40]. Specimen fragmentation remains a significant limitation of EUS-LB because it can significantly compromise diagnostic accuracy [52]. Finally, recent data suggest that EUS-LB with a 19 G FNB needle provides better histologic specimens than does the technique in which FNA needles are used [37] (Figure 1 and Figure 2).

Beyond the needle design and size, there is also the issue of optimal technique to improve the diagnostic yield of EUS-LB. Many endoscopists use suction or slow-pull techniques with FNA. In the human cadaveric study, the type of suction technique did not affect sample adequacy [51]. Hasan et al. used a 22G FNB needle (Acquire; Boston Scientific, Marlborough, MA, USA) and did not apply any suction but developed a technique to limit tissue fragmentation. This technique restricted elevator utilization, and the stylet was slowly re-inserted while keeping the assembly straight to avoid tissue distortion [38]. The wet suction technique, which uses a saline-filled pre-vacuum syringe, showed high effectiveness for EUS-LB, using the 19G Sharkcore or a standard 19G FNA needle even with a single pass and one actuation, as reported in a retrospective study on 165 patients [36]. Furthermore, priming the needle with dilute heparin instead of saline can decrease the formation of blood clots in the needle and improve tissue handling. It has been demonstrated that heparin priming does not lead to bloodier specimens, nor does it increase adverse events of FNA. In a prospective study on 40 patients, using heparin-primed needles improved tissue adequacy compared with dry suction techniques [35]. In a large prospective, multicenter study with a 19G FNA needle, Diehl et al. reported using the fanning technique, a well assessed FNA technique that involves several to-and-fro movements of the needle in the liver with slight variation in the access angle, allowing sampling of new areas of the lobe [29]. The most recent meta-analysis (23 studies, 1326 patients) indicates that using an FNB needle with the slow-pull technique may provide better specimen quality and higher diagnostic yield [53]. Nevertheless, we need more prospective comparative studies to more precisely assess the superiority of various EUS-LB techniques.

There are several limitations to the widespread utilization of this technique. EUS requires a prolonged learning curve to achieve competency [54] in comparison to conventional techniques, which require less technical expertise. Endoscopic equipment and the devices utilized for the procedure are expensive. Conscious sedation or anesthesia further increases the cost, and there are also certain risks in an endoscopic procedure. However, EUS-LB is an evolving technique that already has an important role in settings with relevant expertise, mainly because of the superior control of the operating field, low incidence of adverse events, accessibility of the various parts of the liver, and greater patients’ comfort [55].

### 2.3. EUS-Guided Portal Hypertension Measurement

The hepatic venous portal pressure gradient or portal pressure gradient (PPG) reflects the degree of portal hypertension (PH) and is the best prognostic indicator in liver disease [7]. The current standard for evaluation of PH is an indirect measurement of HVPG via right jugular vein access, where the free hepatic venous pressure is recorded and subtracted from the wedged hepatic venous pressure to determine the HVPG. Despite the overall safety profile, the method is highly invasive and requires technical expertise found at specialized medical centers and may not always accurately reproduce true portal venous pressures, especially in patients with pre- and post-hepatic etiologies of PH [4,56]. Direct percutaneous portal vein catheterization is usually avoided because of the high risk for complications [57]. Due to the relative proximity of the portal vein to the tip of the echoendoscope during the EUS exam, this method emerged as an alternative to standard percutaneous routes for obtaining hepatic vascular access (Figure 1).

The first human case on EUS-guided PPG measurement (EUS-PPG) was reported in 2014 by Fuji-Lau et al. [58]. Subsequently, the prospective pilot study evaluated the use of EUS-PPG measurement in 28 patients with suspected or confirmed cirrhosis by using a 25 G needle and a compact manometer. One hundred percent technical success and no adverse events were reported [59]. A recent study from Zhang et al. confirmed a high degree of consistency between EUS-PPG using a 22-gauge FNA needle and HVPG in patients with acute and subacute PH. The authors showed a strong association between the two variables, with a Pearson correlation coefficient of 0.923 [60]. Although the current literature suggests that EUS-PPG measurement as a means of direct portal pressure evaluation is safe and feasible, larger clinical trials and comparative studies assessing standard methodology and clinical effectiveness of this method are needed. The fact that this can be a part of a multiprocedural intervention (general endoscopic assessment, variceal screening, and EUS-LB and EUS elastography) that can be done during a single endoscopic exam will presumably affect the acceptance of this method in the future [61].

### 2.4. Detection of Varices and Prediction of Variceal Bleeding

Due to the proximity of internal organs to the gastrointestinal tract, one of the main indications of EUS has, so far, been redirected to therapeutic interventions. As a diagnostic tool, EUS has important implications for patients with PH, offering visualization of structures in submucosal spaces, which include varices in the gastrointestinal tract and vascular structures that surround the gastrointestinal wall, and risk evaluation of future bleeding from EGV. The instrument channel in echoendoscopes enables the use of several devices for endoscopic interventions of EGV using glue injection or coil embolization [62].

Variceal hemorrhage is the main cause of upper gastrointestinal bleeding in patients with PH (70% of cases) and represents one of the most serious complications in these patients, with overall six-week mortality at around 15–25% for esophageal varices (EV) and 45% for gastric varices (GV) [63]. According to studies that evaluated the role of EUS in the detection of EV, radial EUS was significantly inferior to standard EGD [4], leaving EGD as an endoscopic method of choice for diagnosis, surveillance, and treatment of EV [64]. However, more recent studies have shown the comparable role of EUS to standard EGD in detecting EV (including small EV), predominantly due to newer and improved technical specifications of echo-endoscopes (smaller tip in echo-endoscope, small water-filled balloons, small 20-Hz ultrasound transducers, high-frequency ultrasound miniature probes, and higher video resolution) [7,8]. EUS can visualize esophageal collateral vessels that can be divided into two main groups: (1) periesophageal collaterals (veins small in size) located close to the esophageal wall; and (2) paraesophageal collaterals (large in size), which are located away from the esophagus [65]. The detection of collateral vasculature that surrounds the esophagus has important clinical implications, predominantly for prognostic purposes. The literature data have shown that the presence of severe collateral and perforation veins detected by EUS can help in the prediction of the recurrence of EV before and after treatment (sclerotherapy or band ligation), suggesting closer follow-up in this subgroup of patients [66,67]. In addition to predicting the risk of variceal recurrence, EUS may also predict the risk of recurrent variceal bleeding after endoscopic variceal ligation (EVL) with sensitivity and specificity around 90% [68]. The most important signs that correlated with higher rates of recurrent variceal bleeding included the diameter of paraesophageal veins and the detection of perforating veins prior to and after endoscopic sclerotherapy, and higher rates of cardiac intramural veins [69,70]. Furthermore, EUS can predict the risk of bleeding by the assessment of the hematocystic spots on the surface of EV (identified as saccular aneurysms), which are closely associated with a high risk of variceal rupture [71]. EV can be eradicated using EUS-guided EVL or sclerotherapy, keeping in mind the abovementioned advantages of EUS in predicting/reducing variceal recurrence. Minor complications after EUS-guided sclerotherapy have been reported, with no significant differences from complications induced by standard EGD [72,73]. A randomized clinical trial that compared standard EGD sclerotherapy and EUS-guided sclerotherapy of the feeding veins to EV showed similar recurrence rates for both groups [73].

### 2.5. EUS-Guided Therapy for PH

On the contrary to EV, gastric varices (GV) are present in a smaller proportion of patients with cirrhosis (20%). It is known that a hemorrhage from cardiofundal varices is less frequent but often more severe and not easily controlled, providing a higher risk of recurrent bleeding and mortality (when compared to EV) [63,74]. While standard EGD still represents the gold standard in detecting EV, EUS has better sensitivity in diagnosing GV [75], with a detection rate two times higher [76]. According to some authors, EUS can evaluate ectopic duodenal varices [77], easily distinguish thickened gastric folds from small GV [78], and help in the diagnosis of portal gastropathy, showing diffuse thickening of the gastric wall with dilated paragastric veins (differential diagnosis to “watermelon stomach”) [7]. EUS also has an important role in the characterization of GV, visualization of treatment in progress, and confirmation of obliteration using Doppler [79,80]. Furthermore, EUS can easily measure the size of GV, which directly correlates with their flow volume [81]. Nowadays, the standard endoscopic management of fundal GV in acute bleeding or selective therapy is endoscopic cyanoacrylate (CYA) injection, which can be complicated with fever, chest pain, post-injection ulcers, re-bleeding (15 to 30%), embolic events (the incidence increases with the amount of CYA injected), or death. One of the advantages of EUS includes the identification of GV in the setting of acute bleeding when blood and clots in the gastric lumen disable an adequate endoscopic view [82]. Since the risk factors for re-bleeding include varix size, presence of para-gastric veins [83], and deficiency of complete obliteration of the GV or of the perforating vascular channels, which are unavailable for detection or eradication during standard EGD, a possible therapeutic role of EUS is arising. A retrospective study on 101 patients treated with glue injection after an episode of GV hemorrhage showed significantly lower re-bleeding rates in those patients in whom EUS was aggressively used during follow-up with the intention of achieving a complete obliteration of variceal veins [84]. EUS-guided hemostasis of GV (with different available methods: injection of CYA, coils, coils with CYA injection, thrombin, or coils with an absorbable gelatin sponge (AGS)) allows assessment of the variceal blood flow, selective targeting of the varices with very exact treatment into the lumen of varix or into its feeding vessel (lowering the required dose of adhesive agent), and monitoring of the obliteration results (confirmation of cessation of variceal blood flow using Doppler and the presence of echogenic GV) [85]. Several studies evaluated the role of CYA injection alone, either in primary prophylaxis [86,87] or acute GV bleeding [88,89,90], with an overall GV obliteration rate of 100% for the first group and 77–100% for the second. The re-bleeding rate was 0% and 5%, respectively, and severe complications were detected in the second group and included pulmonary embolism and splenic infarct in 5% of cases. As it was mentioned earlier, EUS-guided CYA injection has a risk of distal embolization (embolization to the pulmonary arteries and systemic embolism), and a multidisciplinary assessment is required to evaluate the potential presence of a septal defect prior to EUS-guided CYA injection. Furthermore, EUS-guided coil injection (either with or without CYA injection that can be delivered using standard 22- or 19-gauge needles used for FNA) may be a future method of choice to reduce the risk of embolization due to providing primary hemostasis [91,92]. In the literature, limited data showing the role of coil injection alone in treating GV are available. Five studies evaluated coil injection in primary prophylaxis [88,93], both primary prophylaxis and acute GV bleeding [94,95] and in secondary prophylaxis [96]. The results showed a GV obliteration rate of 70–100%, with no re-bleeding complications, but one event of major bleeding during the procedure was detected. Several groups of authors encourage the use of glue injection and coils in combination (primary prophylaxis, acute GV bleeding, and secondary prophylaxis), believing in their synergistic activity of hemostasis and reducing the risk of re-bleeding and distal embolization [85,88,91,96,97,98,99,100]. The overall GV obliteration rate was 40–100%, with up to a 20% re-bleeding rate. A retrospective trial that compared EUS-guided CYA injection to EUS-guided coil placement showed similar rates of varix obliteration (complete obliteration was more likely to be achieved in the coil group after a single endoscopic session) and re-bleeding rates. It was shown that patients treated with CYA injection had significantly higher adverse events, but the number of sessions needed was fewer in patients receiving coil embolization [88]. Some of the adverse events associated with coil placement (with or without CYA injection) include abdominal pain, fever, minor and major bleeding, coil migration, and extrusion of coils into the gastric lumen [88,94,96]. Severe complications also included pulmonary embolism in up to 25% of cases [85,98]. All of the abovementioned results are summarized in Table 2 [85,86,87,88,89,90,91,93,94,95,96,97,98,99,100]. In addition to synthetic tissue adhesives, such as CYA, some of the biologic tissue adhesives that have been studied for GV obliteration include thrombin (converts fibrinogen to fibrin and promotes clot production) and AGS that is prepared from purified porcine gelatin and can absorb up to 45 times its weight in whole blood. The studies showed that EUS-guided thrombin injection had no procedure-related complications, making it safe to use in this indication [101]. EUS-guided coil placement followed by AGS injection is a well-tolerated procedure, providing positive results in small case series [102,103]. EUS-guided CYA injection with/without coiling has also been used for duodenal varices [104]. Despite the abovementioned results, the specific role of EUS-guided coil/CYA injection in primary prophylaxis of EGV is not clear yet. Based on the available data, the treatment strategy should imply aggressive retreatment of any residual GV seen on follow-up EUS, with the intention of achieving their complete obliteration. It is advocated that EUS-guided coil and CYA injection have the best efficacy in the treatment of GV. Due to the previously mentioned advantages of coil placements, EUS-guided coil insertion has been given a preference over CYA injection. In one single-center study, a retrospective cohort of patients with active/recent bleeding or high-risk GV treated with direct endoscopic injection was compared with a prospective cohort of similar patients treated with EUS-guided fine needle injection (EUS-FNI). It was concluded that EUS-FNI is the preferred treatment strategy, which is substantiated by results showing decreased rates of bleeding in the EUS-guided CYA injection group of patients with active or recently bleeding GV [49,90]. According to retrospective analysis that compared patients who underwent EUS-guided coil injection with patients who underwent a standard EGD injection of CYA for secondary prophylaxis of GV, the EUS group had a significantly lower rate of re-bleeding [97]. In conclusion, EUS does not have an established role in clinical practice to investigate PH yet. The only distinct indication for EUS-guided treatment is the failure of standard EGD in GV bleeding control [4,82]. In the future, EUS might provide an alternative approach to transjugular intrahepatic portosystemic shunts in cases of refractory ascites and refractory variceal bleed, and more studies are needed before its eventual implementation in humans.

## 3. EUS in Diagnostic Evaluation of Focal Liver Lesions

### 3.1. EUS FNA/FNB

EUS provides an intimate view of liver anatomy with excellent proficiency in determining the location, size, relation to surrounding structures, and characteristics of focal lesions [4] (Rimbas et al.). Transgastric and transduodenal approaches are used to visualize the left and right liver lobe, respectfully, bearing in mind that the right posterior segments are inaccessible for examination using this method [105]. The value of EUS in detecting and providing tissue diagnosis for liver lesions, most prominently metastases of solid tumors, as well as primary hepatic malignancies, has been shown in multiple studies [4] (Table 3) [106,107,108,109,110,111,112,113,114,115,116,117,118,119,120,121].

A recent review by Sbeit et al. [122] reports the diagnostic yield of EUS-guided biopsy of focal liver lesions ranging from 89.7–100%, confirming the superiority of EUS sampling. The rate of adverse events for tissue sampling under EUS-guidance was reported at 2.3%, including duodenal perforation and death [4]. This rate compares favorably to percutaneous imaging-guided liver tissue acquisition, where the rate of bleeding complications was up to 2% in most studies [123]. Both FNA and FNB have been used for the sampling of focal liver lesions with excellent accuracy, but there are no prospective studies directly comparing the two methods in focal liver lesions. Data coming from studies comparing FNA and FNB in pancreatic lesions suggest a slight superiority of FNB with newer needles requiring fewer passes with higher histological quality [119,124]. Obtaining a high-quality specimen for histopathology is becoming increasingly important for precision medicine in oncology [125]. The need for genetic tumor profiling will probably push in the direction of greater use of 22G needles in the future.

There are several reports highlighting the accuracy of EUS compared to CT in the ability to detect a higher number of neoplastic liver lesions, thus influencing staging and further treatment. However, as noted by Lange et al. [126], most of those studies were performed more than a decade ago and might not reflect the advances in CT and MR modalities. In conclusion, EUS contributes to the staging of hepatobiliary malignancies, but its primary role is tissue acquisition.

### 3.2. EUS Elastography

Elastography uses stiffness quantification to discriminate hard from soft lesions, as the former is usually associated with malignancy. It is an adjunctive method used to better characterize lesions of interest. The use of elastography in EUS has mostly focused on the pancreas with a paucity of data for focal liver lesions. In a review by Lisotti et al., only several studies evaluating the use of elastography in focal liver lesions were identified and are limited by small patient numbers [127]. The use of a hue-histogram cutoff of 170 is associated with a 92.5% sensitivity and an 88.8% specificity. In summary, malignancies are significantly harder than benign lesions and surrounding liver tissue, but further studies are warranted.

### 3.3. Contrast-Enhancement EUS (CE–EUS)

The use of contrast agents for the examination of focal liver lesions allows differentiation based on their microvascular supply and architecture. Owing to the dual blood supply of the liver, ultrasound contrast agents allow examination of focal liver lesions in the arterial, portal, and venous phases. Different patterns of enhancement and contrast washout allow the examiner to better define the nature of the lesion [128]. Minaga et al. found that contrasting harmonic EUS using a second-generation contrast agent (Sonazoid) increased accuracy for the detection of liver metastases of the left lobe up to 98.5% compared to B-mode EUS (91.1%) and CT scan (90.5%). In 6 patients out of 338, only contrasting harmonic EUS could identify metastases impacting staging and further management. In addition, FNA has an excellent accuracy (93.3%) in lesions smaller than 10 mm detected using contrast agents [117]. The use of contrast agents in routine clinical practice adds some complexity but obviously has the potential to significantly augment the detection, characterization, and sampling of focal liver lesions, which is why further studies are needed.

## 4. Staging of Pancreatobiliary Malignancies

Both extrahepatic and intrahepatic cholangiocarcinoma can be detected using EUS and, compared to imaging methods, may provide a more accurate assessment of hilar lesions, vasculature, and distal extrahepatic biliary tree. Perihilar cholangiocarcinoma is associated with a poor prognosis and low resectability rates, but using a model based on a combination of EUS and CT could predict candidates for curative surgery [129]. The European Society for Medical Oncology (ESMO) guidelines recommend using EUS for lymph node staging in biliary cancer, as well as providing information on vessel involvement [130]. Accuracy of EUS for local tumor staging, lymph node involvement, and portal vein infiltration is 66–81%, 64–81%, and 88–100%, respectively [131].

Staging of hepatocellular carcinoma (HCC) is primarily performed using CT and MR, but several studies have shown superiority in detecting small liver lesions and lymph node involvement, potentially changing treatment decisions in a number of patients [132].

Pancreatic cancer is best evaluated using a combination of imaging methods and EUS, which is also used for tissue acquisition. The EUS may provide additional information on staging, especially N-staging, as well as portal vein involvement with less sensitivity for mesenteric and coeliac arteries [133]. The precision and usefulness of EUS in evaluating small lesions have been established in the pancreas, where pancreatic cancer lesions as small as 7 mm have been detected and sampled [134].

Bile duct dilation (BDD) of unclear etiology is a common indication for EUS for concern of malignancy. Differential diagnosis includes pancreatic head cancer, bile duct cancer, ampullary cancer, and chronic pancreatitis, among others. In a systemic review by Smith et al., the cause of BDD was identified in a third of cases and included common bile duct stones, chronic pancreatitis, and periampullary diverticulum, as well as malignancies [135]. A meta-analysis from Sadeghi et al. showed the pooled sensitivity and specificity of EUS-FNA for diagnosis of malignant biliary stricture were 80% and 97%, respectively [136]. Another study demonstrated the superiority of EUS for the detection of malignancies in the setting of biliary stricture compared to CT and MRI (94%, 30%, and 42%, respectively) [137].

In a recent paper by Phan et al., a diagnostic yield of 69.4% was achieved for BDD. Malignancy was confirmed in 8.1% of patients with BDD and normal liver function tests warranting EUS examination in these cases [138].

In conclusion, EUS in conjecture with imaging methods is an indispensable tool in the detection and staging of hepatobiliary malignancies, as well as elucidating obstructive biliary pathology.

## 5. EUS-Guided Treatment of Hepatic Lesions

### 5.1. Treatment of Cystic Liver Lesions

Hepatic cyst ablation is indicated when symptomatic, usually due to size and location. Percutaneous ethanol ablation is an established method, but EUS can be used for cysts located in the left liver lobe. A study comparing the two methods found excellent and durable responses to cyst ablation using ethanol lavage, regardless of the approach used. The advantage of EUS is the one-step approach [139].

### 5.2. Drainage of Liver Abscesses

Liver abscess drainage is commonly performed using a percutaneous approach as the method of choice. This approach is associated with an excellent technical success rate and avoidance of surgery. External drainage and self-removal of the drainage tube are the main drawbacks of this approach. The use of EUS in the drainage of hepatic abscesses is still evolving and mostly based on case reports [140,141,142,143,144,145,146,147,148,149,150,151]. The main advantages are internal drainage, the one-step procedure, and the ability to reach locations inaccessible to the percutaneous approach. Both plastic and self-expanding metal stents can be used, but data supporting a preference of either option is limited. Both percutaneous and EUS approaches are comparable in success and relapse rates, as well as safety [152]. A recent review identified 15 studies describing difficult-to-access liver abscess drainage under EUS guidance followed by stent placement via a guidewire with a reported technical success rate of 97.5% [153]. The transgastric route was commonly used to reach abscesses located in the caudate and left lobe. No major complications were found, and this approach was found feasible and safe for abscesses not accessible by percutaneous drainage.

### 5.3. Treatment of Solid Liver Lesions

EUS can be used for the treatment of focal liver lesions using a variety of techniques described in the literature. This is a relatively new and evolving field, and the available data consist mostly of case series and animal studies. Most of these techniques have been in use via a percutaneous approach, while the use of EUS might provide access to hard-to-reach tumor localizations. Considering the complexities of selecting and managing this type of patients married with the lack of real-world data, clinical application of EUS-guided therapy of solid liver lesions will, for now, be reserved for specialized expert centers as part of a multidisciplinary approach.

#### 5.3.1. FNI Therapy

##### Ethanol Injection Therapy

Targeted ethanol injection of solid liver lesions has, to date, mostly been described for HCC cases located in the caudate or left lobe, with excellent technical success but varying (30–100%) rates of therapeutic response [154,155,156,157,158]. Both 22G and 25G FNA needles have been used for the procedures without major adverse events. The largest reported trial included 26 patients randomized into receiving EUS-guided ethanol injection (*n* = 10) or iodine-125 seed brachytherapy (*n* = 13) for malignant left-sided liver tumors. The treatment achieved a complete response in 65.2% of patients and partial response in 34.8%, and the conclusion was that iodine-125 brachytherapy was superior to ethanol injection with a good safety profile.

Only two case reports of liver metastasis treated with this approach have been published, both originating from pancreatic adenocarcinoma. Technical success was achieved with minor adverse events in both cases, but the lack of any recent publication on this type of treatment effectively limits it to highly individualized cases as an adjunctive palliative treatment modality for patients unsuitable for systemic oncological therapy [159,160].

##### EUS-Guided Portal Injection Chemotherapy (EPIC)

Only preclinical data from porcine models are available for this approach reserved for bilateral hepatic metastases. Irinotecan-loaded beads are injected into the portal vein resulting in an increased hepatic concentration while reducing systemic exposure and adverse effects [161,162].

#### 5.3.2. Thermoablative Therapies

Radiofrequency ablation (RFA), cryothermy, neodymium:yttium-aluminum-garnet (Nd-YAG) laser and high-intensity focused ultrasound ablation have all been tested in animal models for their effectiveness in inducing necrosis of solid tumor lesions of the pancreas and liver [163,164,165,166,167].

##### Clinical Application of RFA Has So Far Been Noted Only in Case-Reports with Fair Success

Jiang et al. performed the largest available study using the Nd-YAG laser for ablation in seven cases of HCC and three cases of colorectal cancer metastases located in the left or caudate liver lobe. Complete response was noted in all cases, but the follow-up was restricted to 3 months. The authors recommend using this method in selected cases not eligible for surgery or percutaneous treatment due to severe comorbidity or advanced CLD. The advantages of this method include a lower complication rate than RFA with a more clearly marked ablation area and without damaging the surrounding tissue [167].

#### 5.3.3. EUS-Guided Brachytherapy and Fiducial Placement

The only study to report treatment of left-sided liver tumor using iodine-125 seed brachytherapy was conducted by Jiang et al., where it was compared to percutaneous ethanol injection as reviewed above and deemed superior [154]. The placement of fiducial markers using EUS with the goal of enabling stereotactic body radiation therapy was shown to be feasible in two retrospective studies [168,169].

#### 5.3.4. Photodynamic Therapy

This novel approach consists of the systemic infusion of photosensitizer material, which accumulates in tumorous tissue and is then activated by optic fiber, resulting in tissue ablation. Several human studies in treating pancreatic tumors have been carried out, but only animal data is available for the treatment of solid liver lesions [170].

## 6. EUS in Primary Sclerosing Cholangitis and Undetermined Biliary Strictures

Primary sclerosing cholangitis (PSC) is a rare chronic cholestatic liver disease that usually affects men. It can occur in different parts of the biliary tree and can cause liver cirrhosis, as well as cholangiocellular carcinoma. Diagnostic work-up of patients with suspected PSC is often challenging. Morphological changes of the biliary tree and liver parenchyma in the early stages of PSC are nonspecific, as well as laboratory parameters. In terms of diagnosis and staging of PSC, magnetic resonance cholangiopancreatography (MRCP), endoscopic retrograde cholangiopancreatography (ERCP), and LB are used [171,172]. On the other hand, the usefulness of EUS in the context of PSC diagnosis is not well investigated. One of the first studies that evaluated the usefulness of EUS in PSC diagnosis was published in 2006 by Mesenas S et al. [173] with promising results in terms of EUS use in PSC diagnosis. Similar results were published a few years later by Croatian authors [174]. More recently, Lutz HH et al. [171] analyzed 138 patients with cholestasis. Of these, 32 patients with possible PSC were evaluated further. In addition to other methods for PSC detection, EUS was included in the diagnostic work-up [171]. The authors evaluated a few parameters: irregular wall structure, wall thickening (≥1.5 mm), significant changes of the caliber of the common bile duct, and perihilar lymphadenopathy. The authors found that EUS had a sensitivity and specificity of predicting PSC of 76.4% and 100%, with positive and negative predictive values of 100% and 79%, respectively [171]. Today, MRCP is the primary diagnostic method in patients with suspected PSC. On the other hand, in patients with early PSC, MRCP findings can be false-negative. In addition, MRCP is not the best method for extrahepatic bile duct visualization. In terms of suspected cholestatic liver disease, EUS should be considered to be added to the diagnostic algorithm, as it is a good method of excluding pancreatic pathologies, such as autoimmune pancreatitis or lithiasis that can be a cause of possible cholestatic disease. On the other hand, strictly intrahepatic PSC or autoimmune cholangiopathy represents a diagnostic gap because these areas of the biliary tree cannot be visualized with the current EUS technology [171]. However, the data we have are few, and we need further multicentric studies that will investigate the role of EUS in the evaluation of suspected PSC.

Undetermined biliary strictures are often a difficult-to-solve clinical dilemma to gastroenterologists. Three years ago, a meta-analysis by De Moura DTH et al. [175] that analyzed EUS and ERCP for tissue diagnosis of malignant biliary stricture was published. In this analysis, 294 patients were included. The authors found that EUS-FNA was a superior method in comparison to the ERCP for the diagnosis of malignant biliary strictures. On the other hand, this method had low negative predictive values; thus, if they are negative, they cannot exclude the malignant etiology of the strictures [4,175]. The second issue in this context is hilar cholangiocarcinoma. According to the data, EUS is useful for the evaluation the nature of the hilar lesion, as well as for providing information on the extent of the periductal disease. In addition, with the help of EUS, the presence of lymph node metastases can be evaluated. In a retrospective study published 13 years ago, authors analyzed the usefulness of EUS-FNA in regional lymph-node staging in a population of 47 patients that had unresectable hilar cholangiocarcinoma awaiting liver transplantation [176]. With the help of EUS, the authors identified lymph nodes in all patients. EUS-FNA confirmed malignant lymph nodes in 17% of patients. Furthermore, the authors noticed that EUS-FNA missed metastatic nodal involvement in two patients. Thus, the overall sensitivity of EUS-FNA was 80% [176].

## 7. Limitations

Limitations to the widespread utilization of this technique are mostly related to cost-effectiveness issues. This drawback is offset in patients who are undergoing an EUS for another indication, for example, LB, after exclusion of biliary pathology in patients with an undetermined liver lesion. There is also a small but non-negligible risk of endoscopy, especially in those with anatomical alteration of the gastrointestinal tract. While the left lobe of the liver can be approached with EUS easily through the gastric wall, accessibility of the right hepatic lobe is limited. Finally, a clear limitation of the current literature of EUS in liver diseases is that the majority of the studies have been small single cohort, single-center, retrospective, and non-randomized [49].

## 8. Conclusions

EUS is one of the most versatile methods in gastroenterology, allowing immediate access with detection, sampling, and treatment of digestive tract pathology. This review summarizes the growing role of EUS in hepatology, consistently showing safe and reliable diagnostic and promising therapeutic potential across various studies. It remains for us to wait for the results of the larger, well-designed, multicentric, and randomized controlled studies to position the role of EUS in diagnostic and therapeutic algorithms in hepatology.

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
