# Peer review of "Role of Endoscopic Ultrasound in Liver Disease: Where Do We Stand?"

_diagnostics, 2021, doi:10.3390/diagnostics11112021_

Round 1
Reviewer 1 Report
This is a review of the usefulness of EUS and Interventional EUS for liver disease. This paper was written after searching through a large number of papers, and is of great significance. I would like to make the following comments. Major 1. The paper is composed of a wide variety of items, which may be difficult for the reader to understand. First, it should be divided into observation-only EUS (including elastography and contrast-enhancement EUS) and interventional EUS including EUS-FNA/B. Next, the techniques should be divided, and then the results for each target lesion should be described. Minor 1. Did you also use "EUS-FNA" or "EUS-FNB" as a keyword in the literature search? Or, did the authors pick up all the articles that correspond to these keywords by using the keywords listed in the articles? 2. There is a noticeable tendency to repeat definitions of abbreviated terms (e.g., "LB", "RTE"). You should check the entire paper and avoid repeating definitions of abbreviations. 3. The word "Also" is often found at the beginning of sentences. However, this is not an appropriate word for a medical article. It should be changed to "In addition" or something similar. 4. The table is written vertically. Since there are many items to be described, it should be changed to horizontal description.
Author Response
Dear reviewer 1,
we would like to thank you for your time and effort invested in reading our article, and above all, for the exhaustive and useful comments. Our responses are outlined below.
Major 1.
The paper is composed of a wide variety of items, which may be difficult for the reader to understand. First, it should be divided into observation-only EUS (including elastography and contrast-enhancement EUS) and interventional EUS including EUS-FNA/B. Next, the techniques should be divided, and then the results for each target lesion should be described.
Response: the primary idea of this article was to synthesize recent insights into the benefits of EUS in the diagnosis and treatment of liver disease. With its growing role in hepatology, we thought about how to structure a large amount of data to make it as easy as possible to follow, but at the same time to give readers an idea about most relevant topics at the moment. With this in mind, and the fact that endohepatology is a hot topic, most of the article is devoted to all aspects of the role of EUS in diffuse liver disease, including diagnosis and therapy of CLD complications. In this way we wanted to unite this part and additionally emphasize the significant (new) role of the EUS in diffuse liver disease. In the next chapter we proceeded with focal lesions, first with the diagnostic role of EUS (with special consideration for staging of pancreatobiliary malignancies) and finally with the role of EUS in therapy of focal liver lesions. We believe that the organization of the article in this form serves well the purpose of informing readers about current trends in this area, although we agree that it is a matter of preference. We fear that a significant change in the structure of the article would affect the message we wanted to convey, especially in light of the comments of other two reviewers who had no objections of this kind.
Minor 1.
Did you also use "EUS-FNA" or "EUS-FNB" as a keyword in the literature search? Or, did the authors pick up all the articles that correspond to these keywords by using the keywords listed in the articles?
Response: we used "EUS-FNB" as a keyword also. We added this in the Introduction section.
There is a noticeable tendency to repeat definitions of abbreviated terms (e.g., "LB", "RTE"). You should check the entire paper and avoid repeating definitions of abbreviations.
Response: we corrected that in the manuscript.
The word "Also" is often found at the beginning of sentences. However, this is not an appropriate word for a medical article. It should be changed to "In addition" or something similar.
Response: we corrected that in the manuscript.
The table is written vertically. Since there are many items to be described, it should be changed to horizontal description.
Response: the tables are originaly created in horizontal orientation. We believe this should be corrected in the final editing.
Reviewer 2 Report
This is a well done review. The tables are well organized. No further recommendations.
Author Response
Dear reviewer 2,
we would like to thank you for your time and effort invested in reading our article, and above all, for the positive comments.
Reviewer 3 Report
This article is very well documented about the available function of EUS for observing and treating liver-related disease.
Authors mentions usefulness of EUS to evaluate staging of pancreatobiliary malignancies. However, the usefulness is limited, because the staging is determined by identifying metastatic lesions of not only liver but also lung, peritoneum, bone, brain, etc. Evaluating lymphnode or vasculature is a part of determining stage of cancer.
On the other hand, EUS is the best method to diagnose small pancreatic cancer. Some reports described that EUS contributes to determining stage to reveal metastatic lesions of the liver from pancreatic cancer. However, the function for staging is also limited.
Authors should consider description about staging of pancreatobiliary malignancies.
Author Response
Dear reviewer 3,
we would like to thank you for your time and effort invested in reading our article, and above all, for the exhaustive and useful comments. Our responses are outlined below.
Authors mentions usefulness of EUS to evaluate staging of pancreatobiliary malignancies. However, the usefulness is limited, because the staging is determined by identifying metastatic lesions of not only liver but also lung, peritoneum, bone, brain, etc. Evaluating lymphnode or vasculature is a part of determining stage of cancer. On the other hand, EUS is the best method to diagnose small pancreatic cancer. Some reports described that EUS contributes to determining stage to reveal metastatic lesions of the liver from pancreatic cancer. However, the function for staging is also limited.
Authors should consider description about staging of pancreatobiliary malignancies.
Response: we have further expanded this section to clarify the points mentioned.
Round 2
Reviewer 1 Report
The author has corrected the points I have made. I also respect the author's opinion. Therefore, I have no additional comments to make.
Author Response
Dear reviewer 1,
thank you again for your time and understanding. Your comments are very appreciated.
Reviewer 3 Report
Bile duct dilation by lower bile duct stricture is not a rare condition. The cause includes pancreatic cancer, bile duct cancer, and chronic pancreatitis. EUS contributes to differentiate these diseases, because MRI or CE-CT is enough for the aim. Author is expected to describe this.
Author Response
Dear reviewer 3,
the comment you mentioned is very appreciated. We have expanded the section and added more references on the subject.